# Feasibility of rabbit auricular VX2 tumor model as an experimental model for intra-arterial embolization

**Woosun Choi**[1], **Byung Kook Kwak**[1]*, **Jisung Jung**[1], **Serah Park**[1], **Soon Auck Hong**[2], **Sang Lim Choi**[3]

**1** Department of Radiology, Chung-Ang University Hospital, Chung-Ang University College of Medicine, Seoul, Republic of Korea, **2** Department of Pathology, College of Medicine, Chung-Ang University, Seoul, Republic of Korea, **3** Department of Radiology, Chung-Ang University Gwangmyeong Hospital, Seoul, Republic of Korea

* kwakbk@cau.ac.kr

**Data Availability Statement:** All relevant data are within the manuscript.

**Funding:** The author(s) received no specific funding for this work.

## Abstract

This study aimed to evaluate the feasibility of VX2 tumor in rabbit auricles as an experimental model for intra-arterial embolization. This study was approved by our Institutional Animal Care and Use Committee. VX2 tumors were implanted in both auricles of 12 New Zealand White Rabbits. To investigate the angiographic and pathologic characteristics, angiography, ultrasonography, and thermography were performed 1–2 weeks after inoculation, and image analysis was performed. The animals were sacrificed thereafter, and histologic analysis was conducted by a pathologist. Tumors did not grow in 3/24 auricles of 12 rabbits used in the experiment, and one rabbit died during anesthesia for an ultrasonographic examination. Therefore, images were obtained from a total of 19 auricles. Angiography was successfully performed in 11 rabbits, and hypervascularity and tumor feeding vessels were clearly observed in all tumors. The enhancement effect increased significantly as the volume increased. In histologic evaluation, the average area of necrosis was 27%. In conclusion, a rabbit auricular VX2 tumor model is easy to create, and its features can be conveniently observed on visual inspection. Moreover, it appears hypervascular on angiography, and the tumor feeding vessel is easy to approach. Thus, it is useful for studying intra-arterial embolization.

## Introduction

The rabbit VX2 tumor model, developed by Rous et al. in the 1930s [1, 2], plays an important role in animal experiments for neoplastic disease. Given the characteristics of VX2 tumors, such as growth rate, hypervascularity, and larger vessels, they are frequently used as animal models for interventional radiology [3]. An experimental animal model created by injecting a VX2 tumor into a rabbit liver is widely used to study the effects of embolization in hepatocellular carcinoma [4–8]. However, researchers have used VX2 tumor cells by injecting them into various organs to create the desired tumor model, which requires invasive surgery. In addition,

**Competing interests:** The authors have declared that no competing interests exist.

it is difficult to visually confirm whether the VX2 tumor grows properly, and thus, additional imaging studies are inevitably necessary to observe its growth. Rabbits characteristically have large auricles with thick arteries and veins; therefore, there are several advantages to creating a tumor model in rabbit auricles [9–11]. A tumor model can be created by injecting cells using a syringe under the auricle without surgery, and the shape and size of the tumor can be observed without imaging studies [9, 10]. In addition, rabbit auricles have thin skin and only few thick hairs; thus, VX2 tumors in the rabbit auricle can also be analyzed by thermography. There have been study on transarterial embolization using VX2 tumors implanted in rabbit auricles [9]. However, studies detailing the pathological, thermography, and angiography findings of VX2 tumors in rabbit auricles have not been reported. Thus, this study aimed to evaluate the feasibility of VX2 tumor in rabbit auricles as an experimental model for intra-arterial embolization. Towards this goal, the angiographic, thermographic, and pathological features of VX2 tumors in rabbit auricles were investigated.

## Materials and methods

### Experimental animals

This study was carried out in strict accordance with the recommendations in the Guide for the Care and Use of Laboratory Animals of the National Institutes of Health. The protocol was approved by the Committee on the Ethics of Animal Experiments of Chung-Ang University (Protocol Number: 14–0014). All surgery was performed under a mixture of tiletamine/zolazepam and xylazine anesthesia, and all efforts were made to minimize suffering.

Twelve New Zealand White Rabbits, weighing 2.6–2.9 kg, were used for this study. The experiment was conducted in both auricles of each rabbit. The animal experiment was conducted over 6 weeks, and the animal health and behavior were monitored daily, except for weekends, during the experimental period. To minimize pain and suffering, anesthetics were used when performing procedures (e.g., injection) or imaging (e.g., angiography). For anesthesia, a mixture of tiletamine/zolazepam (Zoletil; Virbac, Carros, France) 17 mg/kg and xylazine (Rompun; Bayer Korea, Seoul, Korea) 4 mg/kg was injected intramuscularly. The humane endpoint was determined to be euthanasia immediately after the necessary image analysis was completed. This was because experimental animals would have experienced increasingly severe pain and suffering due to the implanted VX2 tumor. If the endpoint was met, the animal was immediately euthanized within the same day. Only one animal died before meeting euthanasia criteria. All involved researchers completed special training programs of animal care or handling as required by the internal regulations of our institution's animal laboratory.

### Tumor cell preparation and generation in rabbit auricle

The VX2 tumor tissue was transplanted into the thigh muscle of the rabbit and was surgically removed. The extracted tissue was cut and sliced into small pieces using scissors and a blade in phosphate-buffered saline. Thereafter, the tissue was passed through a stainless-steel sieve to obtain a single-cell suspension. The suspension was diluted to 20 mL using normal saline and centrifuged at $100 \times g$ for 5 min (washing). The centrifuged cell was prepared as $1 \times 10^7/1mL$ (i.e., 0.1 ml in a 1-ml syringe). The prepared VX2 tumor cell suspension (0.1 mL) was injected using a 26-G needle into the dorsal aspect of the rabbit auricle and subcutaneously into the distal one-third area, 2 cm medial to the central artery. After 1 or 2 weeks of transplanting the VX2 tumor into rabbit auricles, when the average short axis diameter of the tumors was approximately 10 mm visually (transplanting the tumors into 2 to 4 rabbits at a time), it was used for the experiment.

## Thermography

The hair on the rabbit auricles was removed cleanly using a clipper to prevent it from affecting the results. A thermal blanket was placed on the test table to prevent the table temperature from affecting the test results. In addition, to ensure that the body temperature of the rabbit's head and body in the image were not included, a 3-cm cut was made in a sanitary pad mat to expose only the auricle on the test side and cover the head and torso. IRIS 5000 (MEDICORE, Seong-Nam, Korea) was used for thermography, and a prescan was performed 4–5 times to save the images without distortion of the shape of the auricle and tumor. The temperature of the tumor was measured by locating a region of interest (ROI) in the center of the tumor so that the central auricular artery was not included in the stored image. In addition, to measure body temperature in a place without the tumor, an ROI of the same size and shape was selected on the opposite side from the central auricular artery of the tumor, in the non-tumoral area of the auricle. The difference between the values measured in the two ROIs was then analyzed.

## Ultrasonography

Ultrasonography (US) was performed to measure the tumor length and volume. Measurements with the naked eye are not accurate as errors are generated due to the thickness of the skin of the auricle and the curvature of the tumor. Thus, the tumor length and volume were measured by US using a SonoSite MICRO MAXX (HURSCARE, Bothell, USA) L 5–10 MHz linear probe equipment. Tumor size was recorded when the tumor was the largest while scanning it longitudinally and cross-sectionally. In addition, the tumor volume was calculated with the formula $V = (\pi/6) \times X \times Y \times Z \, cm^3$, which calculated the volume of an oval by measuring the depth of the tumor.

## Angiography

Carotid angiography was performed bilaterally using the AXIOM Artis (SIEMENS, München, Germany) frontal C-arm equipment. All procedures were performed surgically and aseptically. The hair in the right inguinal area was cleanly removed using a clipper, the four limbs were tied to an experimental plastic plate, and the animal was maintained in a supine position to perform the Seldinger method. A skin incision was made to be perpendicular to the direction of the right femoral artery, and the subcutaneous tissue was dissected using curved mosquito forceps. Then, the femoral artery was separated from the femoral vein and nerve to expose it. Thereafter, two 5–0 silk sutures were passed under the proximal and distal parts of the femoral artery, and the distal silk suture was ligated and pulled to the distal side to pull the femoral artery. Simultaneously, the artery was punctured using an 18-G angiocatheter, and the plastic sheath was advanced into it. A 4-Fr sheath was advanced into the femoral artery through a 0.018-inch wire. After applying the vascular sheath to the femoral artery, the right and left carotid arteries were selected using a 4-Fr Cobra catheter (Cook, Bloomington, IN, USA) and a 0.035-inch hydrophilic guidewire (Terumo, Tokyo, Japan). Then, an angiography image was recorded for each carotid artery using the 4-Fr Cobra catheter (Cook, Bloomington, IN, USA).

## Image analysis

Angiography images were transferred to the picture archiving and communication system, and images of the end arterial phase in which the tumor appeared to be maximally contrasted were exported in tagged image file format. For each image, the tumor's enhancement effect was calculated using the Image J program (NIH, Maryland, USA). The enhancement effect was defined as the difference in density between the tumoral and non-tumoral areas,

calculated as a percentage. The enhancement effect in the tumoral area was determined by drawing an ROI using the software to include as much of the tumor as possible and exclude the thick blood vessels to obtain the tumor density. Meanwhile, an ROI was drawn on the opposite side from the central auricular artery of the tumor to obtain the density in the non-tumoral area of the auricle.

### Histology

Six rabbits were used for histologic analysis. The remaining 6 rabbits were not used because they could not be sacrificed for use in subsequent studies. After administering anesthesia, the rabbits were sacrificed by injecting potassium chloride through a vein. Immediately after sacrifice, the tumor site of the rabbit auricle was cut and fixed in 10% formaldehyde. The fixed tissue section was stained using the hematoxylin-eosin staining method. The tissue sections were reviewed by an experienced pathologist, and the degree of necrosis, lymphovascular invasion, cartilage invasion, and epidermal damage were analyzed.

### Statistical analysis

The correlation among tumor long axis length, volume, and enhancement effect was analyzed using Spearman's rank correlation coefficient. Statistical analysis was performed using SPSS software (version 14.0. SPSS, Chicago, IL). Statistical significance was set at $P<0.05$.

## Results

### Radiologic findings

Tumors did not grow in 3/24 auricles in the 12 rabbits used in the experiment, and one rabbit died during anesthesia for an ultrasonographic examination. Therefore, imaging tests were obtained from 19 auricles (Table 1). All 19 auricular VX2 tumors could be well observed with the naked eye (Fig 1).

In thermography, the mean temperature of the transplanted VX2 tumors in the auricle was 31.40 ± 1.09 ˚C, and the mean temperature of the normal auricle was 28.38 ± 1.48 ˚C, with the temperature difference between the two being 3.02 ± 0.79 ˚C (Table 1, Fig 2).

The mean volume of the 19 auricle tumors as measured by US was 1.14 ± 0.98 cm$^3$, and the average length of the long axis was 1.58 ± 0.64 cm. Angiography was successfully performed in 11 rabbits. On carotid angiography, tumor enhancement was well observed in rabbit auricles where the VX2 tumor had grown, and tumor hypervascularity and the tumor feeding vessel were also clearly observed in all tumors (Fig 3). The average density changes in the enhancement effect between the tumoral area and the non-tumoral area was 23.48 ± 9.15%. The Spearman's correlation coefficient rho was 0.313 for the correlation between tumor volume and enhancement effect, and the enhancement effect increased significantly as the volume increased ($P$ = 0.029, Fig 4).

### Histologic findings

A total of 6 rabbits were sacrificed. However, the tumor did not grow in one auricle in two rabbits; thus, histologic features were evaluated in 10 auricles (Fig 5). All tumors were located in the subepidermal space and characterized by nodular growth (Fig 5A). Histologically, necrotic areas were observed in all 10 tumors, with an average proportion of 27% (range, 5–50%) (Fig 5B). Tumor cells demonstrated malignant histologic features of pleomorphism and large nuclei, prominent nucleoli, and mitotic activity (Fig 5C). Epidermal damage, manifested as erosion and ulcer, was observed in three auricles (30%). Cartilage damage was observed in one

**Table 1. Tumor findings of the 19 auricles.**

| Auricle no. | Ultrasonography | | Angiography | | Thermography | |
|---|---|---|---|---|---|---|
| | Long axis diameter (cm) | Volume (cm³) | Density | Enhancement effect (%) | Tumoral area temperature (˚C) | Non-tumoral area temperature (˚C) |
| 1—Right | 1.79 | 1.15 | 122.26 | 31.51 | 31.33 | 27.87 |
| 2—Right | 1.29 | 1.75 | 103.43 | 36.15 | 29.11 | 25.61 |
| 2—Left | 1.97 | 1.69 | 103.28 | 17.76 | 29.60 | 26.35 |
| 3—Right | 0.66 | 0.06 | 94.06 | 10.42 | 30.43 | 28.16 |
| 3—Left | 2.15 | 2.51 | 127.93 | 34.77 | 30.90 | 26.87 |
| 4—Right | 2.08 | 2.14 | 93.40 | 13.93 | 31.11 | 27.18 |
| 4—Left | 1.63 | 0.87 | 105.16 | 18.25 | 30.49 | 26.90 |
| 5—Left | 2.29 | 1.76 | 98.87 | 17.35 | 31.25 | 28.39 |
| 6—Right | 3.52 | 3.78 | 73.80 | 30.49 | 31.98 | 27.44 |
| 6—Left | 0.76 | 0.14 | 61.61 | 26.85 | 31.12 | 27.98 |
| 7—Left | 1.00 | 0.29 | 69.24 | 22.28 | 31.95 | 28.90 |
| 8—Right | 1.57 | 0.96 | 81.30 | 18.70 | 33.18 | 29.97 |
| 8—Left | 1.42 | 0.65 | 90.97 | 45.50 | 32.66 | 30.74 |
| 9—Right | 0.92 | 0.12 | 104.08 | 22.24 | 31.07 | 28.21 |
| 9—Left | 1.30 | 0.70 | 121.96 | 32.05 | 31.50 | 28.29 |
| 11—Right | 2.11 | 1.73 | 115.03 | 26.66 | 32.30 | 29.92 |
| 11—Left | 0.87 | 0.13 | 55.07 | 17.96 | 31.09 | 29.54 |
| 12—Right | 1.45 | 0.70 | 86.33 | 15.48 | 33.34 | 30.45 |
| 12—Left | 1.60 | 0.44 | 79.46 | 15.57 | 32.21 | 30.48 |

*Note: no. 1—Left, no. 5—Right, and no. 7—Right = Tumor did not grow; no. 10 = Died during anesthesia

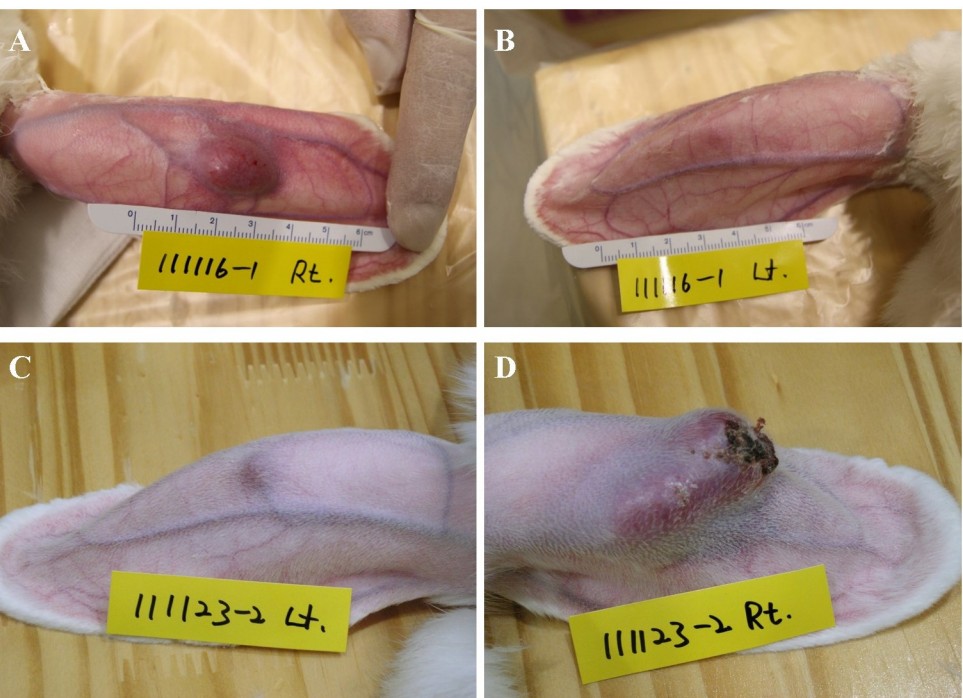

**Fig 1.** Photograph (A) showing a successfully inoculated VX2 tumor in a rabbit auricle Photograph (B) showing a rabbit auricle where the VX2 tumor does not grow. Photograph (C) showing an insufficiently grown VX2 tumor in a rabbit auricle. Photograph (D) showing a successfully inoculated VX2 tumor with a skin ulcer.

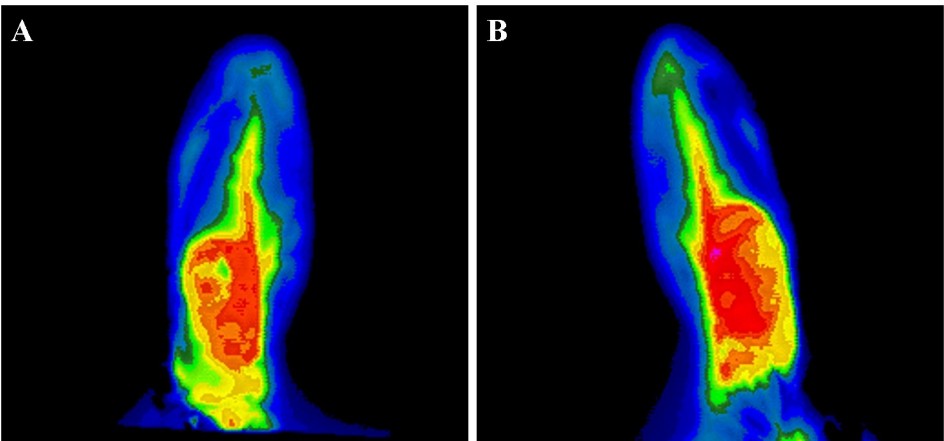

**Fig 2. Thermography images of successfully inoculated VX2 tumors in rabbit auricles (A and B).**

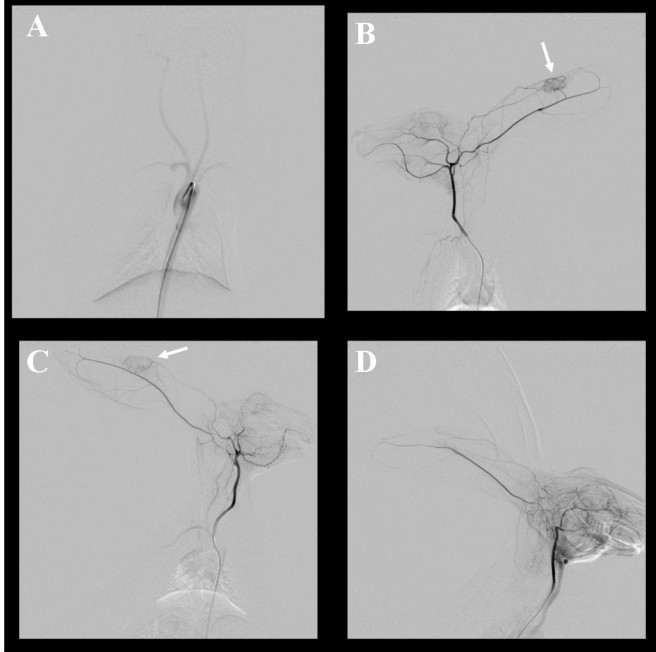

**Fig 3. Carotid angiography via transfemoral arterial access.** Aortography (A) using a 4-Fr catheter shows well-visualized bilateral carotid arteries. Bilateral carotid artery angiography (B and C) shows well-visualized dilated tumor feeding vessels from the central auricular artery and hypervascularity of VX2 tumor (arrow). (D) Carotid angiography in a normal ear reveals no tumor growth.

auricle, and the tumor was observed in both the dorsal and ventral aspects of the cartilage. Lymphovascular invasion was suspected in 5 of the 10 auricles (50%) (Fig 5D).

## Discussion

The pathological, thermography, and angiography findings of VX2 tumors in rabbit auricle have not been reported. In this study, the tumors grew well in 21 of 24 auricles (87.5%) and

# Enhancement effect

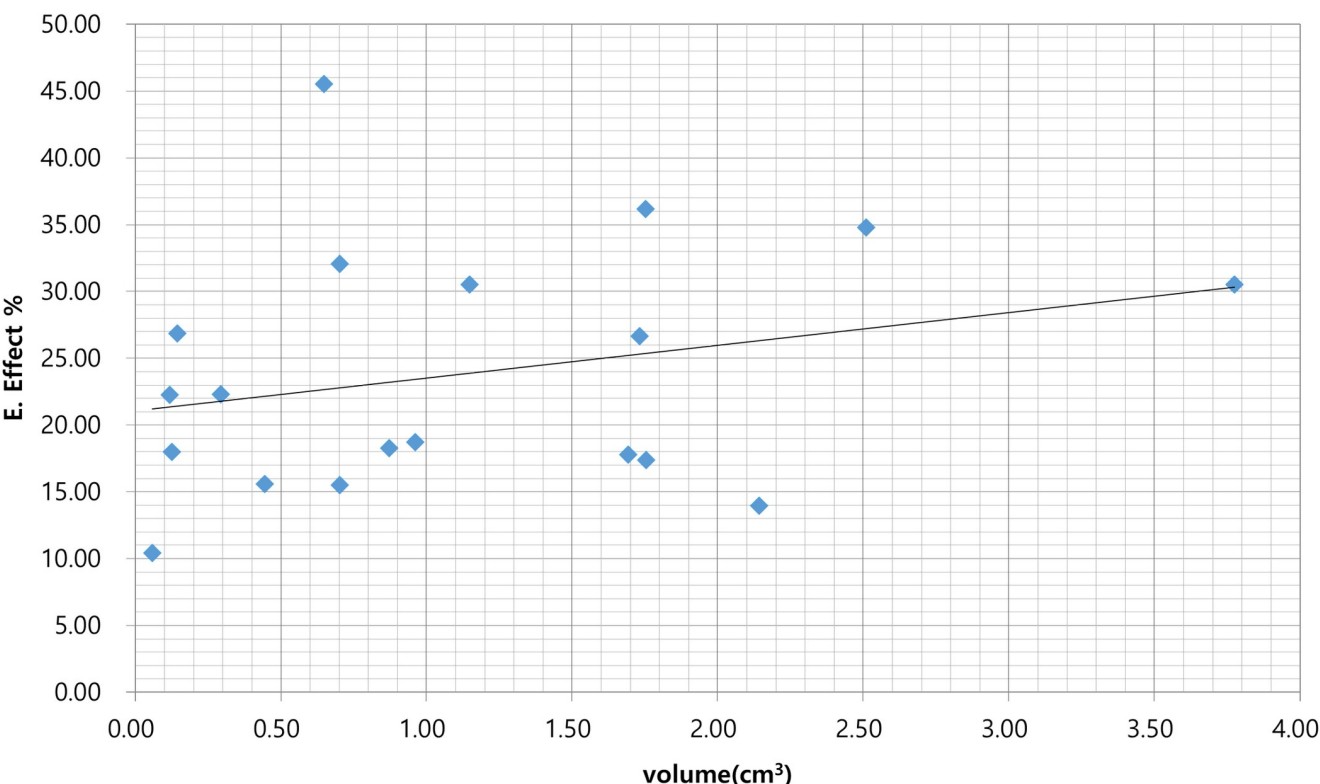

**Fig 4. Correlation between tumor volume and enhancement effect.** The enhancement effect tends to significantly increase as the tumor volume increases (Spearman's correlation coefficient, $P = 0.313$; $P = 0.029$).

were easy to observe with the naked eye. The rate of tumor development in our experiment was not significantly different from that of liver VX2 tumors in other studies [12, 13]. The VX2 tumor model in rabbit auricle is useful in interventional oncology studies. The tumor can be inoculated in the rabbit auricle through a syringe, thus making it more easily accessible. This confers an advantage compared to the need for invasive surgery for transplantation to other organs, such as the liver, uterus, and brain. The auricles are a peripheral structure away from major organs and arteries; thus, tumor growth and manipulation cause less discomfort [11]. In addition, in the VX2 tumor model in rabbit auricle, it was easy to determine the timing for additional experiments because the growth of the tumor could be directly confirmed with the naked eye. In our experiment, as the size of the tumor increased, the enhancement in angiography also significantly increased; thus, it was easy to determine the appropriate timing of the angiography experiment by visual observation. That is, after the tumor model is created, the appropriate timing of the intra-arterial interventional experimental study can be easily determined without additional imaging tests such as US or computed tomography.

In addition, the rabbit auricle has an arterial anatomy suitable for performing angiography [9]. In our study, the tumor feeding vessel could be easily observed and approached during carotid angiography. These characteristics of the VX2 tumor model implanted in the rabbit auricle are advantageous in intra-arterial embolization or infusion studies. In addition, there is no need for additional imaging tests to observe the tumor response after intra-arterial

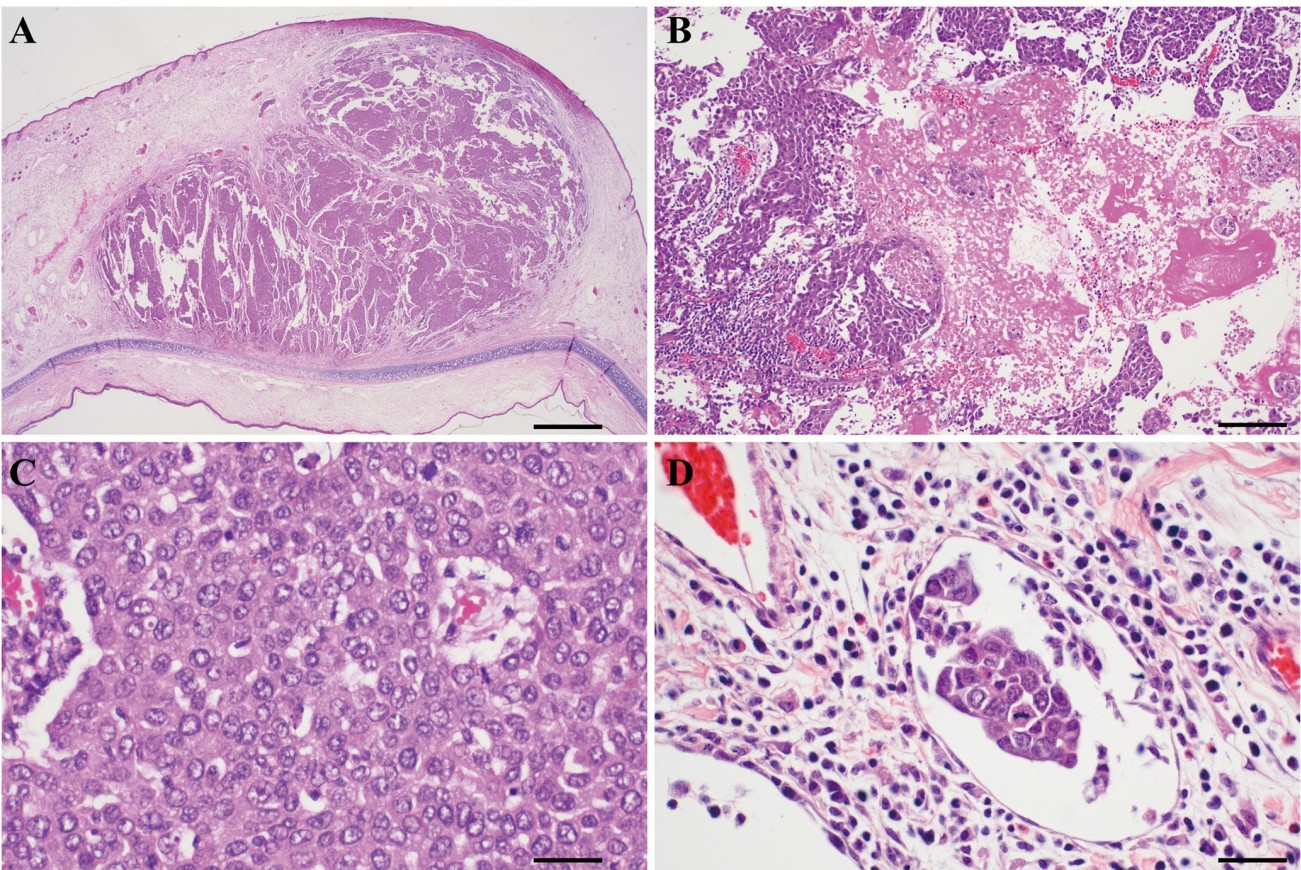

**Fig 5. Microscopy of VX2 tumor in the auricle.** (A) The tumor is located in the subepidermal space and exhibits nodular growth (original magnification, 2×; scale bar, 1000 µm). (B) Areas of necrosis are present (indicated by 'n') (original magnification, 100×; scale bar, 200 µm). (C) High-power magnification reveals malignant tumor cells with pleomorphic, enlarged nuclei and mitotic activity (original magnification, 400×; scale bar, 50 µm). (D) Lymphovascular invasion is suspected (indicated by the arrow) (original magnification, 400×; scale bar, 50 µm).

injection, and the reaction can be observed with the naked eye. Further, the VX2 tumor model in the auricle is also useful in comparative studies between normal and VX2 tumor-implanted auricles. Therefore, this animal model can be used effectively in studies comparing drug or embolic material responses. This would be advantageous not only in intra-arterial injection studies, but also in systemic drug studies. In our study, there was a clear temperature difference between the normal and the VX2 tumor-implanted auricle on thermography. These results suggest that the VX2 tumor model in the rabbit auricle is also suitable for follow-up studies using thermography after drug injection.

Compared to the rabbit hepatic VX2 tumor model, which is commonly used as an arterial embolization model, the rabbit auricular VX2 tumor model has several advantages. First, when preparing the model, the tumor can be more easily accessed by inoculating it into the rabbit ear via a syringe; thus, it is more convenient to prepare this model than preparing the hepatic VX2 tumor model, which generally requires laparotomy; moreover, there is a reduced chance of death of experimental animals due to complications of laparotomy. In addition, the VX2 tumor implanted in the liver requires additional imaging studies, such as CT or US, to determine whether the tumor growth is appropriate for conducting additional experiments such as arterial interventional studies. However, for the auricular VX2 tumor model, the

optimal timing for additional experiments can be easily determined because the growth of the tumor can be directly confirmed with the naked eye. Unlike the liver model, the auricular VX2 tumor model can be visually inspected without additional imaging tests to observe the tumor response after experiments such as intra-arterial embolization or ablation. The VX2 tumor implanted in the liver is commonly used as an intra-arterial embolization model because the blood supply to the tumor from the hepatic artery is very similar to that observed in hepatocellular carcinoma. In particular, the hepatic VX2 tumor model is a hypervascular tumor and the blood supply of the tumor originates in the arterial system [3, 4]. However, although the VX2 implanted in the auricle is supplied from the auricular artery, not the hepatic artery, the tumor originates in the arterial system and has the characteristics of a hypervascular tumor. Therefore, compared to the rabbit hepatic VX2 model, the auricular model is appropriate to use as an arterial embolization model. However, since the surrounding tissues are different, such as the liver and auricle, their characteristics as an experimental model may be different when conducting an ablative study.

Compared to that in other organs, the VX2 tumor model in the rabbit auricle was easier to excise to obtain pathological tissue sections. Pathologically, the current study observed moderate tumor necrosis (average, 27%), supporting that the model is suitable for additional intra-arterial embolization studies. VX2 tumor invasion of the cartilage and epidermis could also be quickly confirmed pathologically. The VX2 tumor in rabbit auricles is pathologically appropriate for intra-arterial embolization studies, and surgical resection to obtain a pathological section after treatment is also easier than in other organs. The VX2 tumor model is widely used in the field of radiologic imaging, especially for liver tumors, and is also important in animal experiments in the field of interventional oncology [6, 14–20]. In addition, the rabbit VX2 model has been useful not only for intra-arterial embolization experiments but also for percutaneous thermal ablation and irreversible electroporation experiments [21–23]. In our study, the femoral artery was selected for the intra-arterial approach. The transfemoral approach is a traditional model for angiography in the rabbit model [3–5, 15]. However, in the rabbit VX2 tumor model, a transauricular approach using the central auricular artery instead of the femoral artery was introduced by Karnabatidis et al. [24]. In addition, previous studies have reported the advantages of the transauricular approach over the transfemoral approach [25]. If the transauricular approach is applied to our experimental model, it may become an easier intra-arterial embolization VX2 tumor model; additional future research is needed.

The present study has some limitations. First, only a small number of rabbits was used, thus limiting the generalizability of the study findings. Therefore, further studies involving a larger number of rabbits will be needed. Second, only six of the 12 rabbits were used in the histologic analysis. However, there are no expected significant differences in results between histologic analysis of six and 12 rabbits. Third, we only performed angiography, and embolization using embolic material was not performed. Therefore, the response of the VX2 tumor in the rabbit auricle to embolization could not be evaluated. We are accordingly planning additional experiments to perform intra-arterial embolization using embolic material.

In conclusion, a VX2 tumor model in rabbit auricles is easy to create, and the features can be conveniently evaluated by visual observation. In addition, it is hypervascular on angiography, and the tumor feeding vessel can be easily approached; hence, it can be used for intra-arterial embolization studies. Given that treatment results can be evaluated without imaging studies, it can be a great advantage in percutaneous injection and intra-arterial embolization experiments. Furthermore, surgical resection can be easily performed; thus, it can be used for pathological examination. We are planning additional experiments to evaluate the response of VX2 tumors in rabbit auricles according to the embolic material using trans-arterial

embolization. It is expected that these additional experiments will prove the effectiveness of the rabbit auricular VX2 tumor model.

## Acknowledgments

We would like to thank Editage (www.editage.co.kr) for English language editing.

## Author Contributions

**Conceptualization:** Woosun Choi, Byung Kook Kwak.

**Data curation:** Woosun Choi, Byung Kook Kwak, Jisung Jung, Serah Park, Soon Auck Hong, Sang Lim Choi.

**Formal analysis:** Woosun Choi, Jisung Jung, Serah Park, Soon Auck Hong.

**Investigation:** Woosun Choi, Byung Kook Kwak, Jisung Jung, Serah Park, Soon Auck Hong, Sang Lim Choi.

**Project administration:** Woosun Choi, Byung Kook Kwak, Jisung Jung, Serah Park.

**Supervision:** Byung Kook Kwak, Jisung Jung, Serah Park, Soon Auck Hong.

**Writing – original draft:** Woosun Choi.

**Writing – review & editing:** Woosun Choi, Byung Kook Kwak, Sang Lim Choi.

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
