## [Decision Letter · Decision Letter 0]

29 Aug 2024

PONE-D-24-24136Feasibility of rabbit auricular VX2 tumor model as an experimental model for intra-arterial embolizationPLOS ONE

Dear Dr. Kwak,

Thank you for submitting your manuscript to PLOS ONE. After careful consideration, we feel that it has merit but does not fully meet PLOS ONE’s publication criteria as it currently stands. Therefore, we invite you to submit a revised version of the manuscript that addresses the points raised during the review process.

According to our reviewers' comments, a Major Revision was needed for this manuscript. Please carefully revise it.

We look forward to receiving your revised manuscript.

Kind regards,

Zhengwei Huang

Academic Editor

PLOS ONE

Journal Requirements:

1. When submitting your revision, we need you to address these additional requirements. Please ensure that your manuscript meets PLOS ONE's style requirements, including those for file naming. The PLOS ONE style templates can be found at https://journals.plos.org/plosone/s/file?id=wjVg/PLOSOne_formatting_sample_main_body.pdf and https://journals.plos.org/plosone/s/file?id=ba62/PLOSOne_formatting_sample_title_authors_affiliations.pdf 2. We note that your Data Availability Statement is currently as follows: "All relevant data are within the manuscript and its Supporting Information files." Please confirm at this time whether or not your submission contains all raw data required to replicate the results of your study. Authors must share the “minimal data set” for their submission. PLOS defines the minimal data set to consist of the data required to replicate all study findings reported in the article, as well as related metadata and methods (https://journals.plos.org/plosone/s/data-availability#loc-minimal-data-set-definition). For example, authors should submit the following data: - The values behind the means, standard deviations and other measures reported;- The values used to build graphs;- The points extracted from images for analysis. Authors do not need to submit their entire data set if only a portion of the data was used in the reported study. If your submission does not contain these data, please either upload them as Supporting Information files or deposit them to a stable, public repository and provide us with the relevant URLs, DOIs, or accession numbers. For a list of recommended repositories, please see https://journals.plos.org/plosone/s/recommended-repositories. If there are ethical or legal restrictions on sharing a de-identified data set, please explain them in detail (e.g., data contain potentially sensitive information, data are owned by a third-party organization, etc.) and who has imposed them (e.g., an ethics committee). Please also provide contact information for a data access committee, ethics committee, or other institutional body to which data requests may be sent. If data are owned by a third party, please indicate how others may request data access. 3. Please include captions for your Supporting Information files at the end of your manuscript, and update any in-text citations to match accordingly. Please see our Supporting Information guidelines for more information: http://journals.plos.org/plosone/s/supporting-information.

Additional Editor Comments:

According to our reviewers' comments, a Major Revision was needed for this manuscript. Please carefully revise it.

Reviewers' comments:

Reviewer's Responses to Questions

**Comments to the Author**

1. Is the manuscript technically sound, and do the data support the conclusions?

Reviewer #1: No

Reviewer #2: Yes

Reviewer #3: Yes

2. Has the statistical analysis been performed appropriately and rigorously? 

Reviewer #1: Yes

Reviewer #2: I Don't Know

Reviewer #3: Yes

3. Have the authors made all data underlying the findings in their manuscript fully available?

Reviewer #1: No

Reviewer #2: Yes

Reviewer #3: Yes

4. Is the manuscript presented in an intelligible fashion and written in standard English?

Reviewer #1: Yes

Reviewer #2: Yes

Reviewer #3: Yes

5. Review Comments to the Author

Reviewer #1: rabbit auricular VX2 tumor model is much easier to establish, comparing with hepatic vx2 tumor model, however, rabbit auricular VX2 tumor model cannot simulate the dual blood supply system of the human, moreover the author should add the regular hepatic vx2 tumor model for the whole comparison. eventually less than half (11/24) models are successful, the feasibility is very low.

Reviewer #2: Feasibility of rabbit auricular VX2 tumor model as an experimental model for intraarterial embolization

The authors provide a well written manuscript describing a small study characterizing VX2 tumor development and characterization in rabbit auricles. The authors characterize growth and characteristics through thermography, ultrasonography, angiography and histopathology, which the authors report has not been previously done for rabbit auricular VX2 tumor research. The majority of the paper is sound. There are some minor deficiencies in the histopathology images and descriptions supplied for this manuscript. I am recommending minor edits which are below.

Thank you to the authors for your curiosity and hard work in generating this manuscript.

Line 51: remove the word ‘height’

Line 102: Within parentheses, change ‘two-four’ to ‘2 to 4’ to be consistent with the authors’ chosen enumeration rather than spelling of single digits. Same with spelled numbers on lines 159, 216, 222, 224

Figure 5 legend: A. remove the word ‘clearly’

C. Remove the word ‘clearly’, remove the phrase ‘abundant mitosis’ and replace with ‘mitotic activity’. Mitoses are most definitely not abundant in the image supplied.

D. I am not convinced that the image depicts lymphovascular invasion. First, image quality is poor. Second, the surrounding fibrovascular tissue somewhat conforms to the purported embolus, and it’s margins around the embolus are fragmented, suggesting artifactual separation from the neoplastic cells. Third, endothelium cannot be seen lining the colorless space. I suspect this is simple tumoral extension with artifactual separation of the underlying stroma.

Please prove me wrong by submitting a higher quality image for figure 5D that shows an endothelial lining, or provide an image of an immunohistochemical stain or other marking examination that highlights the endothelial cells and proves the neoplastic cells are in a vessel.

Reviewer #3: The author implanted VX2 tumors on the auricle of rabbits, and the study has certain significance. However, this manuscript needs to solve the following problems:

1.H&E figure needs to add ruler;

2.Previous studies have reported that VX2 tumors were implanted on the liver (PMID:35667125、36510170). What are the advantages of auricle tumor compared with VX2 tumor implantation in liver?

3. The P in statistical analysis is recommended to be capitalized italics.

6. PLOS authors have the option to publish the peer review history of their article (what does this mean?). If published, this will include your full peer review and any attached files.

Reviewer #1: No

Reviewer #2: No

Reviewer #3: **Yes: **Yanqiao Ren

---

## [Author Response · Author response to Decision Letter 0]

7 Nov 2024

Dear Editor:

We thank you and the reviewers for your thoughtful suggestions and insights. The manuscript has benefited from these insightful suggestions. 

We have answered all the questions below to the best of our ability.

Reviewer Comments to Author:

Reviewer #1

rabbit auricular VX2 tumor model is much easier to establish, comparing with hepatic vx2 tumor model, however, rabbit auricular VX2 tumor model cannot simulate the dual blood supply system of the human, moreover the author should add the regular hepatic vx2 tumor model for the whole comparison. eventually less than half (11/24) models are successful, the feasibility is very low.

-Thank you for your advice. The VX2 tumor is an anaplastic squamous cell carcinoma and is not of hepatocyte origin; hence, it receives only arterial supply and not portal supply when implanted in the liver. This has been described in reference number 4 in our manuscript. The hepatic VX2 model is similar to hepatocellular carcinoma in that the blood supply to the tumor comes from the arterial system; thus, it is a hypervascular tumor and is richly revascularized from the arterial system after treatment. These characteristics are also present in the rabbit auricular VX2 tumor model. However, since we agree that it is important to compare the rabbit liver VX2 model with the auricle VX2 model, we have added a paragraph about the comparison in the discussion section. 

Main text

Compared to the rabbit hepatic VX2 tumor model, which is commonly used as an arterial embolization model, the rabbit auricular VX2 tumor model has several advantages. First, when preparing the model, the tumor can be more easily accessed by inoculating it into the rabbit ear via a syringe; thus, it is more convenient to prepare this model than preparing the hepatic VX2 tumor model, which generally requires laparotomy; moreover, there is a reduced chance of death of experimental animals due to complications of laparotomy. In addition, the VX2 tumor implanted in the liver requires additional imaging studies, such as CT or US, to determine whether the tumor growth is appropriate for conducting additional experiments such as arterial interventional studies. However, for the auricular VX2 tumor model, the optimal timing for additional experiments can be easily determined because the growth of the tumor can be directly confirmed with the naked eye. Unlike the liver model, the auricular VX2 tumor model can be visually inspected without additional imaging tests to observe the tumor response after experiments such as intra-arterial embolization or ablation. The VX2 tumor implanted in the liver is commonly used as an intra-arterial embolization model because the blood supply to the tumor from the hepatic artery is very similar to that observed in hepatocellular carcinoma. In particular, the hepatic VX2 tumor model is a hypervascular tumor and the blood supply of the tumor originates in the arterial system [3, 4]. However, although the VX2 implanted in the auricle is supplied from the auricular artery, not the hepatic artery, the tumor originates in the arterial system and has the characteristics of a hypervascular tumor. Therefore, compared to the rabbit hepatic VX2 model, the auricular model is appropriate to use as an arterial embolization model. However, since the surrounding tissues are different, such as the liver and auricle, their characteristics as an experimental model may be different when conducting an ablative study.

The low feasibility may be due to a calculation error in the title of Table 1 of the main text, which should actually be “19 auricles” but was incorrectly written as “24 rabbits.” We have corrected this in the revised manuscript. In our experiment, the model succeeded in 19 of 24 auricles, except 5, thus indicating a success rate of 79.2%.

Reviewer #2

Feasibility of rabbit auricular VX2 tumor model as an experimental model for intraarterial embolization

The authors provide a well written manuscript describing a small study characterizing VX2 tumor development and characterization in rabbit auricles. The authors characterize growth and characteristics through thermography, ultrasonography, angiography and histopathology, which the authors report has not been previously done for rabbit auricular VX2 tumor research. The majority of the paper is sound. There are some minor deficiencies in the histopathology images and descriptions supplied for this manuscript. I am recommending minor edits which are below.

Thank you to the authors for your curiosity and hard work in generating this manuscript. 

Line 51: remove the word ‘height’

Line 102: Within parentheses, change ‘two-four’ to ‘2 to 4’ to be consistent with the authors’ chosen enumeration rather than spelling of single digits. Same with spelled numbers on lines 159, 216, 222, 224

- We greatly appreciate your comment. We have revised the manuscript according to your comment.

Figure 5 legend: A. remove the word ‘clearly’

C. Remove the word ‘clearly’, remove the phrase ‘abundant mitosis’ and replace with ‘mitotic activity’. Mitoses are most definitely not abundant in the image supplied.

- In accordance with your recommendation, we have removed the word “clearly” and revised the phrases in the main text and figure legends.

Fig 5. Microscopy of VX2 tumor in the auricle (A) The tumor is located in the subepidermal space and exhibits nodular growth (original magnification, 2×; scale bar, 1000 μm). (B) Areas of necrosis are present (indicated by 'n') (original magnification, 100×; scale bar, 200 μm). (C) High-power magnification reveals malignant tumor cells with pleomorphic, enlarged nuclei and mitotic activity (original magnification, 400×; scale bar, 50 μm). (D) Lymphovascular invasion is suspected (indicated by the arrow) (original magnification, 400×; scale bar, 50 μm).

Main text

A total of 6 rabbits were sacrificed. However, the tumor did not grow in one auricle in two rabbits; thus, histologic features were evaluated in 10 auricles (Fig 5). All tumors were located in the subepidermal space and characterized by nodular growth (Fig 5A). Histologically, necrotic areas were observed in all 10 tumors, with an average proportion of 27% (range, 5–50%) (Fig 5B). Tumor cells demonstrated malignant histologic features of pleomorphism and large nuclei, prominent nucleoli, and mitotic activity (Fig 5C). Epidermal damage, manifested as erosion and ulcer, was observed in three auricles (30%). Cartilage damage was observed in one auricle, and the tumor was observed in both the dorsal and ventral aspects of the cartilage. Lymphovascular invasion was suspected in 5 of the 10 auricles (50%) (Fig 5D).

Figure 5 legend: 

D. I am not convinced that the image depicts lymphovascular invasion. First, image quality is poor. Second, the surrounding fibrovascular tissue somewhat conforms to the purported embolus, and it’s margins around the embolus are fragmented, suggesting artifactual separation from the neoplastic cells. Third, endothelium cannot be seen lining the colorless space. I suspect this is simple tumoral extension with artifactual separation of the underlying stroma.

Please prove me wrong by submitting a higher quality image for figure 5D that shows an endothelial lining, or provide an image of an immunohistochemical stain or other marking examination that highlights the endothelial cells and proves the neoplastic cells are in a vessel.

- As discussed with a board-certified pathologist (S.A.H.), in our study, lymphovascular invasion was reanalyzed, and D2-40, CD31, and CD34 immunostainings were performed to confirm lymphovascular invasion. However, while these immunohistochemical stainings were conducted, the relevant area could not be fully evaluated due to its removal during additional sectioning. Although the findings are not definitive, the pathologist suggested a potential lymphovascular invasion based on the following observations: first, the suspected area of lymphovascular invasion was distant from the main tumor; second, there was no surrounding fibrosis; and third, a region suspected to contain endothelial cells was identified. Accordingly, we have revised the main text and figure legend to include the term "suspected." Furthermore, Figure 5d has been updated to display a new image showing the endothelial lining.

Fig 5. Microscopy of VX2 tumor in the auricle (A) The tumor is located in the subepidermal space and exhibits nodular growth (original magnification, 2×; scale bar, 1000 μm). (B) Areas of necrosis are present (indicated by 'n') (original magnification, 100×; scale bar, 200 μm). (C) High-power magnification reveals malignant tumor cells with pleomorphic, enlarged nuclei and mitotic activity (original magnification, 400×; scale bar, 50 μm). (D) Lymphovascular invasion is suspected (indicated by the arrow) (original magnification, 400×; scale bar, 50 μm).

Main text

A total of 6 rabbits were sacrificed. However, the tumor did not grow in one auricle in two rabbits; thus, histologic features were evaluated in 10 auricles (Fig 5). All tumors were located in the subepidermal space and characterized by nodular growth (Fig 5A). Histologically, necrotic areas were observed in all 10 tumors, with an average proportion of 27% (range, 5–50%) (Fig 5B). Tumor cells demonstrated malignant histologic features of pleomorphism and large nuclei, prominent nucleoli, and mitotic activity (Fig 5C). Epidermal damage, manifested as erosion and ulcer, was observed in three auricles (30%). Cartilage damage was observed in one auricle, and the tumor was observed in both the dorsal and ventral aspects of the cartilage. Lymphovascular invasion was suspected in 5 of the 10 auricles (50%) (Fig 5D).

Reviewer #3

The author implanted VX2 tumors on the auricle of rabbits, and the study has certain significance. However, this manuscript needs to solve the following problems:

1.H&E figure needs to add ruler;

- We greatly appreciate your comment. According to your comment, we have added a ruler to the H&E figure.

2.Previous studies have reported that VX2 tumors were implanted on the liver (PMID:35667125、36510170). What are the advantages of auricle tumor compared with VX2 tumor implantation in liver?

- Thank you for your comment. The rabbit auricular VX2 tumor model has the following advantages over the rabbit hepatic VX2 tumor model: the tumor can be more easily accessed by inoculating it into the rabbit ear via a syringe, whereas the liver model generally requires laparotomy. In addition, in the auricular VX2 tumor model, the timing for additional experiments could be easily determined because the growth of the tumor could be directly confirmed with the naked eye. Furthermore, the tumor response can be visually confirmed without additional imaging tests after experiments such as intra-arterial embolization or ablation. We have added a paragraph about the comparison in the discussion section. 

Main text

Compared to the rabbit hepatic VX2 tumor model, which is commonly used as an arterial embolization model, the rabbit auricular VX2 tumor model has several advantages. First, when preparing the model, the tumor can be more easily accessed by inoculating it into the rabbit ear via a syringe; thus, it is more convenient to prepare this model than preparing the hepatic VX2 tumor model, which generally requires laparotomy; moreover, there is a reduced chance of death of experimental animals due to complications of laparotomy. In addition, the VX2 tumor implanted in the liver requires additional imaging studies, such as CT or US, to determine whether the tumor growth is appropriate for conducting additional experiments such as arterial interventional studies. However, for the auricular VX2 tumor model, the optimal timing for additional experiments can be easily determined because the growth of the tumor can be directly confirmed with the naked eye. Unlike the liver model, the auricular VX2 tumor model can be visually inspected without additional imaging tests to observe the tumor response after experiments such as intra-arterial embolization or ablation. The VX2 tumor implanted in the liver is commonly used as an intra-arterial embolization model because the blood supply to the tumor from the hepatic artery is very similar to that observed in hepatocellular carcinoma. In particular, the hepatic VX2 tumor model is a hypervascular tumor and the blood supply of the tumor originates in the arterial system [3, 4]. However, although the VX2 implanted in the auricle is supplied from the auricular artery, not the hepatic artery, the tumor originates in the arterial system and has the characteristics of a hypervascular tumor. Therefore, compared to the rabbit hepatic VX2 model, the auricular model is appropriate to use as an arterial embolization model. However, since the surrounding tissues are different, such as the liver and auricle, their characteristics as an experimental model may be different when conducting an ablative study.

3. The P in statistical analysis is recommended to be capitalized italics.

- We greatly appreciate your comment. According to your comment, we have revised the term to capital italics.

Once again, we express our gratitude for your insightful comments.

---

## [Decision Letter · Decision Letter 1]

9 Dec 2024

Feasibility of rabbit auricular VX2 tumor model as an experimental model for intra-arterial embolization

PONE-D-24-24136R1

Dear Dr. Byung Kook Kwak,

We’re pleased to inform you that your manuscript has been judged scientifically suitable for publication and will be formally accepted for publication once it meets all outstanding technical requirements.

Kind regards,

Zhengwei Huang

Academic Editor

PLOS ONE

Additional Editor Comments (optional):

The manuscript is acceptable.

Reviewers' comments:

Reviewer's Responses to Questions

**Comments to the Author**

1. If the authors have adequately addressed your comments raised in a previous round of review and you feel that this manuscript is now acceptable for publication, you may indicate that here to bypass the “Comments to the Author” section, enter your conflict of interest statement in the “Confidential to Editor” section, and submit your "Accept" recommendation.

Reviewer #1: All comments have been addressed

2. Is the manuscript technically sound, and do the data support the conclusions?

Reviewer #1: Yes

3. Has the statistical analysis been performed appropriately and rigorously? 

Reviewer #1: Yes

4. Have the authors made all data underlying the findings in their manuscript fully available?

Reviewer #1: Yes

5. Is the manuscript presented in an intelligible fashion and written in standard English?

Reviewer #1: Yes

6. Review Comments to the Author

Reviewer #1: The current manuscript is acceptable as all the issues from the reviwers have been well addressed after revision.

7. PLOS authors have the option to publish the peer review history of their article (what does this mean?). If published, this will include your full peer review and any attached files.

Reviewer #1: **Yes: **Qun Tang

---

## [Editor Report · Acceptance letter]

13 Dec 2024

PONE-D-24-24136R1 

PLOS ONE

Dear Dr. Kwak, 

I'm pleased to inform you that your manuscript has been deemed suitable for publication in PLOS ONE. Congratulations! Your manuscript is now being handed over to our production team.

Kind regards, 

on behalf of

Dr. Zhengwei Huang 

Academic Editor

PLOS ONE